# Management of Central Venous Catheters in Children and Adults on Home Parenteral Nutrition: A French Survey of Current Practice

**DOI:** 10.3390/nu14122532

**Published:** 2022-06-18

**Authors:** Julien Gotchac, Florian Poullenot, Dominique Guimber, Emmanuelle Ecochard-Dugelay, Stéphane Schneider, Noël Peretti, Lore Billiauws, Corinne Borderon, Anne Breton, Emilie Chaillou Legault, Cécile Chambrier, Aurélie Comte, Marie-Edith Coste, Djamal Djeddi, Béatrice Dubern, Claire Dupont, Lucile Espeso, Philippe Fayemendy, Nicolas Flori, Ginette Fotsing, Swellen Gastineau, Olivier Goulet, Emeline Guiot, Adam Jirka, Jeanne Languepin, Sabrina Layec, Didier Quilliot, Laurent Rebouissoux, David Seguy, Isabelle Talon, Anne Turquet, Marjolaine Vallee, Stéphanie Willot, Thierry Lamireau, Raphael Enaud

**Affiliations:** 1Pediatric Intensive Care Unit, Children’s Hospital, Université de Bordeaux, 33000 Bordeaux, France; 2Department of Gastroenterology, Haut-Leveque Hospital, Université de Bordeaux, 33600 Pessac, France; florian.poullenot@chu-bordeaux.fr; 3Department of Paediatrics, Division of Gastroenterology, Hepatology and Nutrition, Jeanne de Flandre Children’s Hospital, University of Lille, 59000 Lille, France; dominique.guimber@chru-lille.fr; 4Department of Pediatric Gastroenterology and Nutrition, Robert Debré Hospital, Assistance Publique-Hôpitaux de Paris 75019 Paris, France; emmanuelle.dugelay@aphp.fr; 5Department of Gastroenterology and Clinical Nutrition, Nice University Hospital, 06000 Nice, France; schneider.s@chu-nice.fr; 6Department of Pediatric Gastroenterology-Hepatology and Nutrition, Hôpital Femme Mere Enfant HFME, Hospices Civil de Lyon HCL, 69500 Bron, France; noel.peretti@chu-lyon.fr (N.P.); lucile.espeso@gmail.com (L.E.); 7CarMeN Laboratory, INSERM U1060, INRAE U1397, Université Claude Bernard Lyon-1, 69921 Oullins, France; 8Department of Gastroenterology and Nutrition Support, Beaujon Hospital, Assistance Publique-Hôpitaux de Paris, 92110 Clichy, France; lore.billiauws@aphp.fr; 9Department of Paediatrics, Clermont-Ferrand University Hospital, 63000 Clermont-Ferrand, France; cborderon@chu-clermontferrand.fr; 10Department of Paediatrics, Gastroenterology, Hepatology, Nutrition, and Diabetes, Purpan Hospital, Toulouse University, 31059 Toulouse, France; breton.a@chu-toulouse.fr; 11Department of Paediatrics, Angers University Hospital, 49933 Angers, France; emchaillou@chu-angers.fr; 12Nutrition Intensive Care Unit, Lyon Sud Hospital, Hospices Civils de Lyon, 69495 Pierre-Benite, France; cecile.chambrier@chu-lyon.fr; 13Pediatric Medicine Unit, Besançon University Hospital, 25000 Besançon, France; a2comte@chu-besancon.fr; 14Intestinal Failure Program, Department of Pediatrics, La Timone Hospital, Assistance Publique des Hôpitaux de Marseille, 13385 Marseille, France; marie-edith.coste@ap-hm.fr; 15Department of Pediatrics and Adolescent Medicine, Amiens University Hospital, 80000 Amiens, France; djeddi.djamal-dine@chu-amiens.fr; 16Pediatric Nutrition and Gastroenterology Department, Trousseau Hospital, Assistance Publique-Hôpitaux de Paris, Sorbonne University, 75012 Paris, France; beatrice.dubern@aphp.fr; 17Department of Paediatrics, Caen Normandie University Hospital, 14033 Caen, France; dupont-c@chu-caen.fr; 18Nutrition Unit, Limoges University Hospital, 87042 Limoges, France; philippe.fayemendy@chu-limoges.fr; 19Gastroenterology Department, Montpellier Cancer Institute, University of Montpellier, 34090 Montpellier, France; nicolas.flori@icm.unicancer.fr; 20Gastroenterology Department, La Milétrie Hospital, Poitiers University Hospital, 86000 Poitiers, France; makougang.fotsing@chuv.ch; 21Department of Pediatrics, Rennes University Hospital, 35200 Rennes, France; swellen.gastineau@chu-rennes.fr; 22Division of Paediatric Gastroenterology-Hepatology-Nutrition, Necker Enfants Malades Hospital, Assistance Publique-Hôpitaux de Paris, Descartes University, 75015 Paris, France; olivier.goulet@aphp.fr; 23Department of Pediatrics, Brabois Hospital, Nancy University Hospital, 54511 Vandoeuvre-lès-Nancy, France; e.guiot@chru-nancy.fr; 24Nutrition Support Team, Division of Gastroenterology, Hôtel-Dieu Hospital, Nantes University Hospital, 44093 Nantes, France; adam.jirka@chu-nantes.fr; 25Department of Pediatrics, Limoges University Hospital, 87042 Limoges, France; jeanne.languepin@chu-limoges.fr; 26Nutrition Department, St-Yves Clinic, 35000 Rennes, France; layec@clinique-styves.fr; 27Department of Diabetology-Endocrinology-Nutrition, Brabois Hospital, Nancy University Hospital, 54511 Vandoeuvre-lès-Nancy, France; quilliot.d@orange.fr; 28Department of Pediatric Gastroenterology and Hepatology, Children’s Hospital, Université de Bordeaux, 33000 Bordeaux, France; laurent.rebouissoux@chu-bordeaux.fr (L.R.); thierry.lamireau@chu-bordeaux.fr (T.L.); raphael.enaud@chu-bordeaux.fr (R.E.); 29Service Endocrinologie Diabétologie Maladies Métaboliques et Nutrition, Centre Hospitalier Universitaire de Lille, Université de Lille, 59037 Lille, France; david.seguy@univ-lille.fr; 30Department of Pediatric Surgery, Strasbourg University Hospital, 67200 Strasbourg, France; isabelle.talon@chru-strasbourg.fr; 31Department of Pediatrics, Felix Guyon University Hospital, 97400 Saint-Denis, France; anne.turquet@chu-reunion.fr; 32Department of Hepatology, Gastroenterology, and Nutrition, Caen Normandie University Hospital, 14033 Caen, France; marjolainevallee@hotmail.com; 33Department of Pediatrics, Clocheville Hospital, Tours University Hospital, 37044 Tours, France; s.willot@chu-tours.fr

**Keywords:** chronic intestinal failure, venous thrombosis, central venous catheter thrombosis, catheter obstruction

## Abstract

Although central venous catheter (CVC)-related thrombosis (CRT) is a severe complication of home parenteral nutrition (HPN), the amount and quality of data in the diagnosis and management of CRT remain low. We aimed to describe current practices regarding CVC management in French adult and pediatric HPN centers, with a focus on CVC obstruction and CRT. Current practices regarding CVC management in patients on HPN were collected by an online-based cross-sectional survey sent to expert physicians of French HPN centers. We compared these practices to published guidelines and searched for differences between pediatric and adult HPN centers’ practices. Finally, we examined the heterogeneity of practices in both pediatric and adult HPN centers. The survey was completed by 34 centers, including 21 pediatric and 13 adult centers. We found a considerable heterogeneity, especially in the responses of pediatric centers. On some points, the centers’ responses differed from the current guidelines. We also found significant differences between practices in adult and pediatric centers. We conclude that the management of CVC and CRT in patients on HPN is a serious and complex situation for which there is significant heterogeneity between HPN centers. These findings highlight the need for more well-designed clinical trials in this field.

## 1. Introduction

The prognosis of patients with chronic intestinal insufficiency has greatly improved since the development of home parenteral nutrition (HPN) [1,2]. A central venous access is mandatory for the perfusion of hyperosmolar fluids required to achieve nutritional goals. The main complications of HPN are related to these vascular accesses, the most frequent being infectious and mechanical complications such as occlusion (thrombotic or not), thrombosis, extravasation, breakage, and migration [3]. The presence of a central venous catheter (CVC) is a significant risk factor for thrombosis, especially in children [4]. The incidence of CVC-related deep vein thrombosis (CRT) is not precisely defined in children on HPN, ranging from 1% to 75% in the literature [3,5]. It is better documented in adults with an incidence rate around 0.045 episode/year in the first year following catheter insertion [6]. CRT is associated with significant morbidity, and can lead to loss of venous access, a critical situation for patients on long-term HPN. In children, repermeabilization is observed in only half of CRTs, and a post-thrombotic syndrome develops in 30% of them [4]. In adults, 15 to 36% of CRTs are complicated by pulmonary embolism and can be followed by post-thrombotic syndrome in up to 46% of cases [6].

Venous thrombosis, a relatively common event in adults, has been studied extensively, leading to guidelines based on a strong level of evidence [7]. However, of these, few concern CVC occlusions and CRTs in patients with HPN and randomized clinical trials on this specific issue are scarce in the literature. Nevertheless, the European Society for Clinical Nutrition and Metabolism published guidelines [2,8] on the particular situation of CVC occlusion and CRTs in adult HPN patients.

On the other hand, venous thrombosis is a quite rare event in children [9,10], and has been less investigated in pediatrics. In 2012, the American College of Chest Physicians (ACCP) published guidelines for the management of thrombosis in children [11], which were completed in 2018 by the American Society of Hematology [12]. European guidelines have been recently published more specifically on thrombosis in children on HPN [3]. However, randomized controlled intervention trials are lacking in this field, and these guidelines mostly rely on the extrapolation from adult studies and expert opinion.

The main objective of our study was to look at current practices in the management of CVC via a survey in French HPN centers, especially regarding CVC obstruction and thrombosis. The secondary objectives were to assess the implementation of guidelines and to compare practices between pediatric and adult HPN centers.

## 2. Materials and Methods

Data regarding the management of CVCs and CRT in patients on HPN were collected using an online-based survey proposed to HPN centers, members of the MaRDi (Maladies Rares Digestives), a French network on rare digestive diseases, belonging to the FIMATHO network, itself part of the European Rare Disease ERNICA network. The questionnaire was sent to 31 pediatric centers and 22 adult centers.

The questionnaire was created by a working group of the MaRDi network. After a review of the literature, items were discussed among members of the working group, and modifications were added. After several rounds of discussion, the final version of the questionnaire (see Appendix A) included 24 questions divided into 5 parts: venous access characteristics, patients’ follow-up modalities, management of CVC obstructions, management of the thromboembolic risk, and management of catheter-related venous thrombosis. Although CRTs were our focus, we considered it essential to obtain information on CVC characteristics and management and on CVC obstructions as they are part of the spectrum of catheter thrombosis.

It was tested and timed by two physicians. The response time ranged from 8 to 10 min and no formulation or semantic problems were identified.

The online survey was then released by e-mail in May 2019 to one corresponding physician per each HPN center of the MaRDi network. All correspondents were permanent physicians, designated as referents for HPN in their centers. To maximize the response rate, the online survey remained open for a 10-month period during which 3 reminders were sent out by email.

The results are presented in tables distinguishing answers between adult and pediatric centers. Quantitative variables were described as mean ± standard deviation (SD) when distribution was normal, or median with interquartile ranges (IQR) otherwise. Qualitative variables were described as a percentage of the total population. Qualitative variables were compared between adult and pediatric centers using χ^2^ with correction for continuity, or an exact Fisher’s test if the χ^2^ conditions were not met. Quantitative variables were compared between adult and pediatric centers using the Student’s *t*-test when there was equality of variance between the groups and when they followed a normal distribution. When their distribution did not follow a normal distribution, the Mann–Whitney test was used if variance between groups was equal and the Welch’s test if not. To assess heterogeneity, the rate of consensus among centers was calculated for each of the 34 qualitative questions in the survey. It was considered that a consensus was obtained for an item when the same answer to the question was given by 80% of centers or more. Below 80% of the same answer, it was concluded that there was heterogeneity of practices on this item. This heterogeneity was assessed separately for pediatric and adult centers. Analyses were performed with RStudio version 3.5.0 (RStudio, PBC, Boston, MA, USA) and the Jamovi 1.2 graphical interface (The jamovi project, Sydney, Australia). Differences with a *p* value of less than 0.05 were considered statistically significant.

## 3. Results

### 3.1. Participation Rate

The survey was completed by 34 centers (64% response rate), including 21/31 (68 %) pediatric centers, and 13/22 (59%) adult centers. Of the physicians who completed the survey, seven (21%) had a Ph.D.

### 3.2. Characteristics of Venous Access

The centers’ responses regarding venous access characteristics in patients on HPN are summarized in Table 1.

Tunneled catheters were the only CVC type used by 43% of the pediatric centers, while 57% of the pediatric centers and all the adult centers used various CVC types (*p* = 0.02). In the pediatric centers, tunneled CVCs were inserted by either anesthesiologists (67% of centers) or surgeons (52% of centers). Radiologists were not involved in the pediatric centers, while they were the most frequently cited specialists in the adult centers (77%) (*p* < 0.01). Antibacterial in-line filters were always used in half of the pediatric centers, and never used in 77% of the adult centers (*p* = 0.03). The frequency of use of a catheter clamp or of positive pressure bidirectional valves was not different between the adult and pediatric centers.

As shown in Table 1, there was no consensus among the pediatric centers on any question about venous access characteristics. The adult centers agreed on one (CVC type used).

### 3.3. Patient Follow-Up Modalities

The centers’ responses regarding follow-up modalities of patients on HPN are summarized in Table 2.

Patients were followed-up every 3 months (IQR: 2; 3) in the pediatric centers and every 3.5 months (IQR: 3; 4.9) in the adult centers (*p* = 0.01). Blood sampling was systematically performed via the CVC in 90% of the pediatric centers, whereas 77% of the adult centers preferred peripheral sampling (*p* < 0.01). Most centers provided education to the patient or their caregiver in order to detect complications such as CVC obstruction or CRT. All the centers except three (one pediatric and two adult) reported that they did not systematically change tunneled CVCs in the absence of complications. Routine imaging prescription was not significantly different between the adult and pediatric centers. Systematic annual prescription of an imaging examination was unusual, except for chest X-rays in a third of the pediatric centers. In both adults and pediatrics, magnetic resonance (MR) angiography was almost never prescribed. More information regarding the choice of imaging according to each clinical situation is provided in the Appendix A.

Regarding the questions listed in Table 2, there was a consensus in both the pediatric and adult centers on one question (venous Doppler ultrasound prescription), only in the pediatric centers on one question (blood sampling site), and only in the adult centers on three questions (chest X-ray, computed tomography (CT) and MR angiography use).

### 3.4. Management of CVC Obstruction

The centers’ responses regarding the management of CVC obstruction in patients on HPN are summarized in Table 3.

In the situation of CVC obstruction, no imaging examination was systematically performed by nearly half of the pediatric or adult centers. Of the nine pediatric centers performing imaging, eight cited chest X-ray and four cited venous Doppler ultrasound (d-US). Among the seven adult centers performing routine imaging in this situation, five centers cited d-US and four centers cited chest X-ray. For occluded CVCs, in the case of a failed restoration of patency procedure, 17 pediatric centers (81%) and 9 adult centers (69%) did not immediately change the CVC but attempted several more procedures to restore CVC patency (*p* = 0.43). The median maximum number of additional attempts before changing the CVC was 2 (IQR: 1; 3) in children and 1.5 (IQR: 0; 2) in adults (*p* = 0.12).

Regarding the chosen molecule for restoring CVC patency, other responses than urokinase were as follow: alteplase (five pediatric centers), streptokinase (one adult center), and missing answers (two pediatric centers).

Regarding the three questions listed in Table 3, there was a consensus in both the pediatric and adult centers on one question (protocol for restoring CVC patency) and only in the adult centers on one question (molecule for restoring CVC patency).

### 3.5. Management of the Thromboembolic Risk

The centers’ responses regarding the management of the thromboembolic risk in patients on HPN are summarized in Table 4.

Searching for thrombophilia was rarely performed systematically before HPN onset. Two pediatric centers reported doing so only in the case of an unstable inflammatory disease. After a diagnosis of CRT, it was systematically performed in 71% of the pediatric centers and 23% of the adult centers (*p* = 0.02). In five pediatric centers, the specific indications of thrombophilia searching after a diagnosis of CRT were the presence of other clinical features suggesting thrombophilia (repeated thrombosis, atypical localization, etc.) or the absence of catheter-related bloodstream infection (CRBSI), known as a CRT risk factor. In seven adult centers, the specific indications for thrombophilia searching after a diagnosis of CRT were multiple, extended, or repeated thrombosis, thrombosis without additional risk factors, and ischemic patients.

The diagnosis of thrombophilia resulted in a long-term full dose anticoagulation regimen in 33% of the pediatric centers and 62% of the adult centers (*p* = 0.21).

Of the 21 pediatric centers, 7 (33%) reported performing primary preventive anticoagulation in the case of thrombophilia (4 centers), loss of venous access due to multiple thrombosis (3 centers), and severe inflammatory bowel disease (3 centers). Of the 13 adult centers, only 2 (15%) reported indications for primary preventive anticoagulation in the case of thrombophilia (1 center) and a past history of thrombosis (1 center).

Regarding the five questions about the management of the thromboembolic risk listed in Table 4, there was a consensus in both the pediatric and adult centers on one question (search for thrombophilia before HPN onset) and only in the adult centers on one question (thromboprophylaxis indications).

### 3.6. Management of Catheter-Related Venous Thrombosis

The centers’ responses regarding the management of CVC obstruction in patients on HPN are summarized in Table 5.

Several significant differences between the pediatric and adult centers regarding the management of CRT were observed. First, the specialist questioned for advice (*p* < 0.01) was a cardiopediatrician in 33% of the pediatric centers, but never a cardiologist in the adult centers. The adult centers requested the advice of an angiologist in 62% of cases. Monitoring imaging was performed early after treatment initiation (less than one month, see Table 4) in 52% of the pediatric centers, whereas adult centers performed imaging later on (*p* < 0.01). Asymptomatic thrombosis was systematically treated with anticoagulation in 85% of the adult centers and in only 38% of the pediatric centers (*p* = 0.02). Finally, systematic monitoring of anti-Xa activity was performed in all pediatric centers compared to only 38% of adult centers (*p* < 0.01). Heparin was used as the first-line curative treatment of CRT in all centers. Fifteen pediatric centers (71%) and eleven adult centers (92%) reported using low molecular weight heparin (LMWH) and three pediatric centers (14%) reported using unfractionated heparin. Two pediatric centers (10%) and two adult centers (17%) reported sometimes using a vitamin K antagonist.

Regarding the six questions about the management of CRT listed in Table 4, there was a consensus only in the adult centers on one question (management of asymptomatic thrombosis) and only in the pediatric centers on one question (anti-Xa activity monitoring).

## 4. Discussion

### 4.1. Main Results

The answers to the survey were analyzed for their heterogeneity, their compliance to the current guidelines, and the differences between the pediatric and adult centers. Overall, we found considerable heterogeneity, especially in the responses of the pediatric centers. On some points, the centers’ responses differed from the current guidelines. We also found significant differences between the practices in the adult and pediatric centers.

Despite the heterogeneity, there was a consensus on certain topics in both the adult and pediatric centers. Venous Doppler ultrasound examination was not performed as a routine radiological screening, but rather when CRT was clinically evoked, as recommended by the guidelines [2,13,14]. Nearly all centers reported having a written and validated internal protocol for the restoration of patency of occluded CVCs. The most commonly used drug was urokinase, but a fourth of the pediatric centers cited alteplase, which is the recommended agent in children [3]. Both alteplase and urokinase are mentioned in adult guidelines [8].

Screening for thrombophilia was not routinely performed prior to HPN onset. Indications for thrombophilia screening in HPN patients are not mentioned in HPN guidelines. British clinical guidelines for testing for thrombophilia in adults also states that testing is not recommended in patients with CRT [15]. ACCP pediatric guidelines state that the presence of thrombophilia should not modify deep vein thrombosis management, leading to the absence of screening for it [11].

Adult centers reported no indication for primary thromboprophylaxis. Indeed, the guidelines state that the decision to start preventive treatment in patients with CVCs remains unsupported by evidence [8]. Conversely, adult centers were prone to prescribe anticoagulation for incidentally discovered asymptomatic thrombosis, whereas the guidelines were unable to achieve a consensus on this point.

In children, as recommended [11], pharmacological monitoring of anti-Xa activity was routine practice in patients treated by LMWH.

As recommended, the adult centers preferred peripheral blood sampling, whereas the pediatric centers used CVCs. This difference is explained in the guidelines: although peripheral blood sampling reduces the use of the CVC, and thus potential complications [2], blood sampling via the CVC is preferred in children to improve quality of life [16]. The adult centers, but not the pediatric centers, agreed on imaging prescriptions: chest X-ray and angio-CT were not prescribed systematically. Unlike in pediatrics, an annual chest X-ray to assess the appropriate position of the central line in adults is not recommended [2,13].

### 4.2. Issues in Pediatrics

In the pediatric centers, the answers to questions about characteristics of venous access and CRT management showed the greatest heterogeneity, the least compliance with the guidelines, and the most differences with adult practices. This probably illustrates a lack of data on these fields in pediatrics, due to the scarcity of thrombosis in children [9,10]. Regarding CRT management, this lack of knowledge may be one reason for a more aggressive management than is recommended [3,11] with more frequent: systematic searches for thrombophilia after a first episode of CRT, anticoagulation lasting more than 3 months, indications for long-term curative anticoagulation, and early control of imaging. Regarding venous access characteristics, the pediatric centers were divided about the use of in-line filters: nearly half of them reported routinely using those filters, versus only one adult center, although neither pediatric nor adult guidelines reported evidence to support their use [2,16]. Regarding radiological control after CVC insertion, as in adults, most centers reported using only per-operative image intensification, and did not routinely prescribe postoperative chest X-rays. This strategy is acceptable in adults as long as the catheter is not placed using a blind subclavian approach [8]. Pediatric guidelines are not as explicit, but they specify that the correct position of the CVC tip must be controlled by a chest X-ray [16].

### 4.3. Issues in Adult Centers

To a lesser extent, there were also several areas of disagreement among the adult centers. They were divided on the usefulness of imaging prescriptions for CVC obstruction, which is not surprising since there is no recommendation on this point. Interestingly, a majority of the pediatric centers reported prescribing no imaging in this situation although pediatric guidelines state that “Thrombotic catheter occlusion (…) require thorough investigation” [3]. Although the adult centers prescribed long-term full-dose anticoagulation more often than the pediatric centers, only half of them reported such a practice in the case of thrombophilia or persistent thrombosis. The most recent ASH guidelines do not specifically address CRTs, but state that secondary prevention (i.e., indefinite antithrombotic therapy) should be prescribed after the completion of primary treatment for deep vein thrombosis provoked by a chronic risk factor [7]. It is, however, debatable whether a CVC for HPN can be considered as a significant chronic risk factor.

Primary prevention in patients with thrombophilia is generally not addressed by guidelines which state that thrombophilia testing “may be helpful in patients with venous thromboembolism at intermediate risk of recurrence, in whom the finding of a strong thrombophilia can be one of the arguments for long-term anticoagulation”, hence arguing against primary prophylaxis [17]. On the other hand, ESPEN guidelines state that thromboprophylaxis, but not long-term full-dose anticoagulation, seems reasonable in patients on HPN with a high risk of venous thromboembolism [8]. It can easily be argued that patients with inherited “strong” thrombophilia match this definition.

### 4.4. Limitations and Perspectives

This study has several limitations. First, this cross-sectional study provides a point-in-time picture of current practices and was not designed to assess the efficacy of practices. There may be a participation bias, since the HPN centers most concerned or comfortable with this topic could be more prone to participate in this study. Another possible bias is that, because there was only one responding physician per center, the response may reflect personal practice more than center practice if there is heterogeneity among several physicians from the same center. Moreover, some of the observed heterogeneity may be due to a difference in years of experience in the field among the responding physicians. Finally, this survey may induce a reporting bias, since centers may give answers which conform to the guidelines, rather than reflecting the actual practices in their department.

To interpret our findings, it is important to keep in mind that comparisons between pediatric and adult centers should be made with caution, particularly regarding thrombosis, as there are many differences between these two populations. First, these two populations have a very different physiology of hemostasis [18,19]. Second, adult patients often have additional risk factors for thrombosis. For instance, as mesenteric ischemia is the most frequent cause of short bowel syndrome in adults in France, it may induce a bias towards over prescribing anticoagulants. Third, the ratio of CVC size to vessel diameter is higher in children compared to adults [5], and CVCs are then a more significant risk factor in children.

Our results only give an indication of French HPN center practices, and spreading this survey to the European level, where practices are supposed to be harmonized, could be an interesting perspective.

Nevertheless, this study provides valuable insights about French HPN center practices. Indeed, the participation rate was high and nearly all major HPN centers participated.

Two main considerations can be highlighted from this study. First, the insufficient implementation of the current pediatric guidelines is of concern, and probably due to the low level of evidence of studies in pediatrics and/or the frequent extrapolation of adult data. This had been shown in pediatrics at the European level by Hojsak et al. before the publication of the latest guidelines, who found a great heterogeneity of practice in Europe and a low compliance with the guidelines of that time [20].

Second, it seems necessary to intensify clinical research on certain topics. Indeed, the guidelines and the current literature do not provide answers to certain issues such as the relevance of screening thrombophilia before HPN onset and/or thromboprophylaxis [11,16,21,22,23,24,25] and are sometimes even contradictory, for example regarding the necessity to treat asymptomatic thrombosis in children [3,12].

## 5. Conclusions

This first French national survey on the management of central venous catheters in HPN centers shows that CVC and CRT management practices vary greatly among centers and do not always comply with the current guidelines, especially in pediatric centers. Efforts should be made by the French HPN centers and the MaRDI network to harmonize practices among HPN centers.

## Figures and Tables

**Table 1 nutrients-14-02532-t001:** Venous access characteristics.

Question	Pediatric Centers (n = 21)	Adult Centers (n = 13)	*p*
CVC type			0.02
Tunneled catheter only All types (tunneled, PICC line, implantable port) ^a^	9 (43%)	0 (0%)	
	12 (57%)	13 (100%)	
Tunneled catheter insertion			
Anesthesiologist	14 (67%)	4 (31%)	0.092
Radiologist	0 (0%)	10 (77%)	<0.01
Surgeon	11 (52%)	5 (38%)	0.66
Control after CVC insertion			0.46
Image intensification	12 (57%)	8 (62%)	
Chest X-ray	0 (0%)	1 (8%)	
Both	9 (43%)	4 (31%)	
Use of a clamp			0.33
Always	12 (57%)	7 (54%)	
Sometimes	1 (5%)	3 (23%)	
Never	8 (38%)	3 (23%)	
Use of positive pressure bidirectional valves			0.50
Always	15 (71%)	7 (54%)	
Sometimes	4 (19%)	4 (31%)	
Never	2 (10%)	2 (15%)	
Use of antibacterial filters			0.03
Always	10 (48%)	1 (8%)	
Sometimes	3 (14%)	2 (15%)	
Never	8 (38%)	10 (77%)	

^a^ Consensus in adult centers, CVC: central venous catheter, PICC: peripherally inserted central catheter.

**Table 2 nutrients-14-02532-t002:** Patient follow-up.

Question	Pediatric Centers (n = 21)	Adult Centers (n = 13)	*p*
Patient or caregiver education in detecting complications such as obstruction and thrombosis			0.85
Always Sometimes	14 (67%)	10 (77%)	
	2 (10%)	1 (8%)	
Never	5 (24%)	2 (15%)	
Venous Doppler ultrasound prescription			0.37
Never	1 (5%)	0 (0%)	
Systematic (annual)	3 (14%)	0 (0%)	
On certain indications ^b^	17 (81%)	13 (100%)	
Chest X-ray prescription			0.23
Never	1 (5%)	1 (8%)	
Systematic (annual)	7 (33%)	1 (8%)	
On certain indications ^a^	13 (62%)	11 (85%)	
Angio-CT prescription			0.20
Never	9 (43%)	2 (15%)	
On certain indications ^a^	12 (57%)	11 (85%)	
Angio-MR prescription			0.88
Never ^a^	16 (76%)	11 (85%)	
On certain indications	5 (24%)	2 (15%)	
Blood sampling site			<0.01
Central (catheter) ^p^	19 (90%)	2 (15%)	
Peripheral	2 (10%)	10 (77%)	
Missing answer	0 (0%)	1 (8%)	

^a^ Consensus in adult centers. ^b^ Consensus in both adult and pediatric centers. ^p^ Consensus in pediatric centers. CT: computed tomography. MR: magnetic resonance.

**Table 3 nutrients-14-02532-t003:** Management of CVC obstruction.

Question	Pediatric Centers (n = 21)	Adult Centers (n = 13)	*p*
Protocol for restoring CVC patency ^b^	19 (90%)	13 (100%)	0.69
Imaging prescription			0.868
No	12 (57%)	6 (46%)	
Before the procedure for patency restoration	8 (38%)	6 (46%)	
After the procedure for patency restoration	1 (5%)	1 (8%)	
Molecule for restoring CVC patency			0.20
Urokinase ^a^	14 (67%)	12 (92%)	
Other	7 (33%)	1 (8%)	

^a^ Consensus in adult centers. ^b^ Consensus in both adult and pediatric centers.

**Table 4 nutrients-14-02532-t004:** Thromboembolic risk.

Question	Pediatric Centers (n = 21)	Adult Centers (n = 13)	*p*
Search for thrombophilia before HPN onset			0.76
Systematic Specific indications	1 (5%)	1 (8%)	
	2 (10%)	0 (0%)	
Never ^b^	18 (86%)	12 (92%)	
Search for thrombophilia after a diagnosis of CRT			0.02
Systematic	15 (71%)	3 (23%)	
Specific indications	5 (24%)	7 (54%)	
Never	1 (5%)	3 (23%)	
Thromboprophylaxis indications ^a^	7 (33%)	2 (15%)	0.45
Long term full-dose anticoagulation			
Persistent thrombosis	7 (33%)	7 (54%)	0.41
Thrombophilia	7 (33%)	8 (62%)	0.21

^a^ Consensus in adult centers. ^b^ Consensus in both adult and pediatric centers.

**Table 5 nutrients-14-02532-t005:** Management of catheter-related venous thrombosis.

Question	Pediatric Centers (n = 21)	Adult Centers (n = 13)	*p*
Prescription of anticoagulation			0.07
Protocolized	5 (24%)	2 (15%)	
Systematic specialist advice	14 (67%)	5 (38%)	
Occasional specialist advice	2 (10%)	6 (46%)	
Specialist advice			<0.01
Angiologist Hemostasis specialist	2 (10%)	8 (62%)	
	12 (57%)	2 (15%)	
Cardiologist/cardio-pediatrician	7 (33%)	0 (0%)	
Missing answer	0 (0%)	3 (23%)	
Treatment duration			0.27
<6 weeks	2 (10%)	0 (0%)	
6 weeks–3 months	7 (33%)	9 (69%)	
>3 months	5 (24%)	2 (15%)	
Unspecified	7 (33%)	2 (15%)	
Imaging control			<0.01
Early (≤1 month)	11 (52%)	0 (0%)	
Late (≥6 weeks or after treatment completion)	5 (24%)	8 (62%)	
Unspecified	5 (24%)	5 (38%)	
Management of asymptomatic thrombosis			0.02
Anticoagulation ^a^	8 (38%)	11 (85%)	
No anticoagulation	2 (10%)	0 (0%)	
Request for specialist advice	11 (52%)	2 (15%)	
Systematic anti-Xa activity monitoring ^p^	21 (100%)	5 (38%)	<0.01

^a^ Consensus in adult centers. ^p^ Consensus in pediatric centers.

## Data Availability

Data supporting the reported results can be found at CHU de Bordeaux, Service de Gastropédiatrie, Hôpital Pellegrin, CHU de Bordeaux, 33076 Bordeaux, France.

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
