# Peer review of "Management of Central Venous Catheters in Children and Adults on Home Parenteral Nutrition: A French Survey of Current Practice"

_nutrients, 2022, doi:10.3390/nu14122532_

Round 1

Reviewer 1 Report

In this French survey the authors report data on management of central venous catheter (CVC) and related complications among children and adults on home parenteral nutrition.

The topic is interesting and the manuscript is well written.

To enrich their relevant data the authors should add (if possible) more details on physicians participant to the survey (age, years of experience in the field). This would permit to distinguish between doctor-related biases and center-related biases. 

Author Response

Response to Reviewer 1 Comments

Point 1: To enrich their relevant data the authors should add (if possible) more details on physicians participant to the survey (age, years of experience in the field). This would permit to distinguish between doctor-related biases and center-related biases. 

Response 1: Thank you for your interest in this work. You do raise an important and interesting point, since theses bias are of particular concern in the interpretation of practice analysis questionnaires. 

Unfortunately, we do not have readily available data on the age and years of experience of the respondents to the survey. This is indeed a pitfall and these two questions should have been included in our questionnaire.

However, the survey was completed by physicians designated as referents for home parenteral nutrition in their centers. These are therefore permanent physicians with a significant experience, although it may indeed differ from one physician to another. For example, of the 34 responding physicians, 7 had a PhD. 

We have provided these additional informations in the Materials and Methods section (line 139-140), and in the Results section (line 164-165). The bias that may result from a possible heterogeneity in respondent seniority was also addressed in the Discussion section (lines 352-353).

Reviewer 2 Report

The work presents an interesting aspect related to home nutritional treatment. It gives an overview of the recommendations used and the reason to follow them, as long as clear standards are developed.

Author Response

Response to Reviewer 2

Point 1: The work presents an interesting aspect related to home nutritional treatment. It gives an overview of the recommendations used and the reason to follow them, as long as clear standards are developed

Response 1: Thank you for your interest in this work. We hope that our results will contribute to encourage clinical research in this field.